# From birth to lying on- or under a supplemental heat source: How long does it take for half the litter to lie down and how long do they stay?

**Cassandra R. Stambuk, Anna K. Johnson⊙\*◉, Karli J. Lane, Kenneth J. Stalder◉**

Department of Animal Science, Iowa State University, Ames, Iowa, United States of America

◉ These authors contributed equally to this work.
\* johnsona@iastate.edu

**Data Availability Statement:** All relevant data are within the paper. The original data set can be made available upon request to the corresponding author.

## Abstract

Piglets are susceptible to hypothermia because they lack hair and energy reserves, have a large surface area to body weight ratio, and have poor body thermostability. Different heat sources are used on farm, but it is not known how long it takes half a litter to locate it and lie down. The objectives of this study were to determine 1) how long it took for ≥ 50% of piglets to locate and lie on- or under the supplemental heat source for ≥ 5 min after the birth of the last-born pig and, 2) how long this cohort of piglets laid on- or under the heat source. A total of 12 sows were enrolled in the study (parity 1 = 4; 3 = 2; 4 = 2; 5 = 2; and 7 = 2). The stall containing one sow and her litter was the experimental unit. Two treatments were compared: 1) Baby Pig Heat Mat—Single 48 (**MAT**) and 2) Poly Heat Lamp (**LAMP**). Temperature was 32˚C for both heat source treatments. Sow and litter video recordings occurred continually over a 24-h period on the day of farrowing. Two measures were determined 1) how long it took for ≥ 50% of piglets to locate and lie on- or under the supplemental heat source for ≥ 5 min after the birth of the last-born pig (h:min), and 2) how long this cohort of piglets laid on- or under the heat source (min:sec). Lying was defined as either sternal or lateral recumbency with ≥ 75% of the piglet's body touching the heat mat or inside the lamp heat circle. Production records were used to verify farrowing date, total number of piglets born, and born alive. No cross fostering occurred during this study. All data will be presented descriptively. On average, sows assigned to the LAMP treatment took ~2 ½-h to farrow, and for sows assigned to the MAT ~3 ½-h, respectively. Piglets took between ~5-h (LAMP) and ~9-h (MAT) for ≥ 50% of piglets to locate and lie on- or under the supplemental heat source for ≥ 5 min after the birth of the last piglet. Cohort of piglets laid on- or under the heat source as follows, LAMP piglets spent ~29 mins lying and for MAT piglets ~42 mins, respectively. Average pre-weaning mortality was 11% (LAMP) and 18% (MAT). The MAT heat source used less energy than the LAMP (16 vs. 63 kWh) over the study duration. To the authors' knowledge, this is the first published study using a continuous sampling method to precisely examine a new measure (time needed for ≥ 50% of piglets to locate and lie on- or under the supplemental heat source for ≥ 5 min after the birth of the last-born piglet) and to determine how long this cohort of piglets laid on- or under the heat source. Our findings show an

**Funding:** This project was funded in part by Iowa Farm Bureau Federation (AJ) and Kane Manufacturing (KS). Partial funding of Dr. Johnson's and Dr. Stalder's salary is supported by the Department of Animal Science, College of Agriculture and Life Sciences at Iowa State University, and the US Department of Agriculture. The funders had no role in study design, data collection and analysis, decision to publish or preparation of the manuscript.

**Competing interests:** The authors have declared that no competing interests exist.

immense range in locating and lying under- or on the heat source. Therefore, we suggest that caretakers should assist all piglets to locate the heat source after farrowing is complete to improve piglet livability.

## Introduction

Satisfying the different thermal requirements for sows and piglets is a challenge for caretakers. The optimal thermal range for piglets is between 32 to 35°C, whilst for the lactating sow the optimal thermal range is 15 to 26°C [1]. Piglet pre-weaning mortality has been well researched, with contributing factors including, gender [2], birth weight [3], vitality [4], and hypothermia [5]. Piglets are susceptible to hypothermia because they lack hair and energy reserves, have a large surface area to body weight ratio, and have poor body thermostability [6, 7]. Important secondary effects of hypothermia are starvation and death resulting from piglets huddling too close to their dam for warmth [8, 9]. To reconcile the sow and piglet thermal differences, caretakers provide piglets supplemental heat sources [9, 10]. Heat sources typically utilized on commercial farms include radiant heating (lamps) or conductive heating (mats). Heat lamps are a low initial investment cost. Wattage (both infrared and incandescent) bulbs range between 100 to 250 W. Electric heat mats are more energy efficient at 60 to 110 W [10–13].

Heat source type, placement, and the number of piglets using the heat source during the first day post-farrowing has been evaluated [9, 10, 14]. Zhu et al., [15] reported that piglets spent more time on water heated mats than on electrically heated mats and under heat lamps between birth and weaning.

Training caretakers with practical Animal Based Measures (ABM) is critical for neonatal management to improve survivability. These ABM's may include knowing the time interval between piglet births and determining if assistance is required to drying piglets. What would be valuable is to design and test a novel on-farm practical ABM to improve neonatal survivability related to the important heat source in the farrowing area (*personnel communication*). If caretakers can determine how quickly, or not piglets seek and use the heat source, they can play a more proactive role in helping some piglets seek warmth and, in turn improve the chances of survival. However, there is no evidence in the scientific literature examining how long it takes for half of the litter to navigate to- and lie on- or under the heat source for the first time or how long this cohort rests. The objectives of this study were to determine 1) how long it took for $\geq 50\%$ of piglets to locate and lie on- or under the supplemental heat source for $\geq 5$ min after the birth of the last-born pig and, 2) how long this cohort of piglets laid on- or under the heat source.

## Materials and methods

The research protocol was approved by the Iowa State University Institutional Animal Care and Use Committee (8-17-8583-S). Sows or piglets were all cared for following the approved farms standard operating procedures. No sows or piglets were anaesthetized, provided analgesics or euthanized during the study.

### Location and housing

The study was conducted at the Iowa State University Allen E. Christian Swine Teaching Farm in Ames, IA. Each sow was provided her own farrowing stall. Farrowing stalls had interlocked plastic flooring and a creep area on both sides of the sow measuring 2.0 m x 0.6 m. The total

stall area measured 2.0 m x 1.7 m and the center sow area measured 2.0 m x 0.6 m. Solid floor-ing was provided on one side of the piglet creep area, where the heat source provided mea-sured 1.2 m x 0.4 m. The stalls were distributed across two farrowing rooms in a negative pressure, mechanically ventilated barn where the temperature was set at 21˚C [9, 10]. Sows were provided 72-h acclimation period to the farrowing stall with the heat source treatment turned on prior to farrowing. All sows and piglets were cared for by the farms Standard Oper-ating Procedures.

## Treatments

A total of 12 sows were enrolled in the study (parity 1 = 4; 3 = 2; 4 = 2; 5 = 2; and 7 = 2). The stall containing one sow and her litter was the experimental unit. Two treatments were com-pared: 1) Baby Pig Heat Mat—Single 48 (**MAT** n = 6; 34 cm W x 122 cm L with a heating area of 0.4 m$^2$, polyethylene; Kane Manufacturing, Pleasant Hill, IA; Fig 1A) and 2) Poly Heat Lamp Fixture (**LAMP** n = 6; 25 cm L x 30 cm W with a heating area of 0.3 m$^2$, polypropylene; HogSlat, Newton Grove, NC; Fig 1B) with a 125 W Infrared Heat Bulb (QC Supply, Ames, IA).

Temperature was 32˚C for both heat source treatments and this was confirmed using a handheld infrared temperature gun (Tool House Digital Infrared Thermometer: model 770343S, Alltrade Tools, LLC, Long Beach, CA; ± 2˚C) held directly under the lamp and the outer perimeter of the light circle, and at each mat edge and center. The LAMP was controlled via a single-step mechanical thermostat for a maximum temperature, while the MAT was con-trolled via Thermostat Programmable 13 Zone (Kane Manufacturing, Pleasant Hill, IA).

## Behavioural evaluation

Sow and litter video recordings occurred continually over a 24-h period on the day of farrow-ing. Video was recorded using a 12 V color Close Circuit Television (CCTV) camera (Model WV-CP484, Matsushita Co Ltd., Japan) positioned centrally 2.5 m above each stall. Behavioral measures were captured digitally utilizing a Noldus portable lab (Noldus Information Tech-nology, Wageningen, The Netherlands). Cameras were connected into a multiplexer, allowing images to be recorded using a PC with HandiAvi (v4.3, Anderson's, AZcendant Software, Tempe, AZ, USA) at 30 fps.

During training, one person with 2-yrs of behavioral experience reviewed all of the video. A ≥ 95% intra-reliability was achieved. The obser watched all farrowing videos in real time on VLC media player (v3.0.10, VideoLan). Video was viewed to ascertain when the first and last piglet was born. This provided the total time for farrowing (h:min).

Two measures were determined: 1) how long it took for ≥ 50% of piglets to locate and lie on- or under the supplemental heat source for ≥ 5 min after the birth of the last-born pig (h: min) and 2) how long this cohort of piglets laid on- or under the heat source (min:sec). Lying was defined as either sternal or lateral recumbency with ≥ 75% of their body touching the heat mat or inside the lamp heat circle [9, 10]. Production records were used to verify farrowing date, total number of piglets born, and born alive. No cross fostering occurred during this study. All data will be presented descriptively.

## Results

On average, sows assigned to the LAMP treatment took ~2 ½-h to farrow, and for sows assigned to the MAT ~3 ½-h, respectively. Piglets took between ~5-h (LAMP) and ~9-h (MAT) for ≥ 50% of piglets to locate and lie on- or under the supplemental heat source for ≥ 5 min after the birth of the last-born pig. There was no consistent pattern related to sow

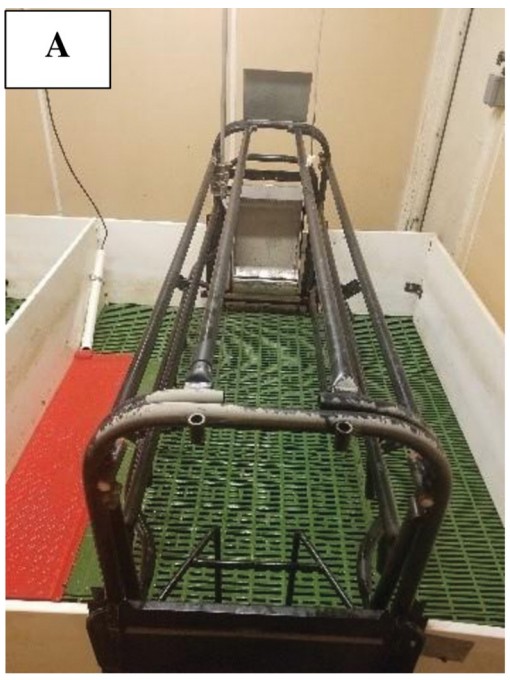

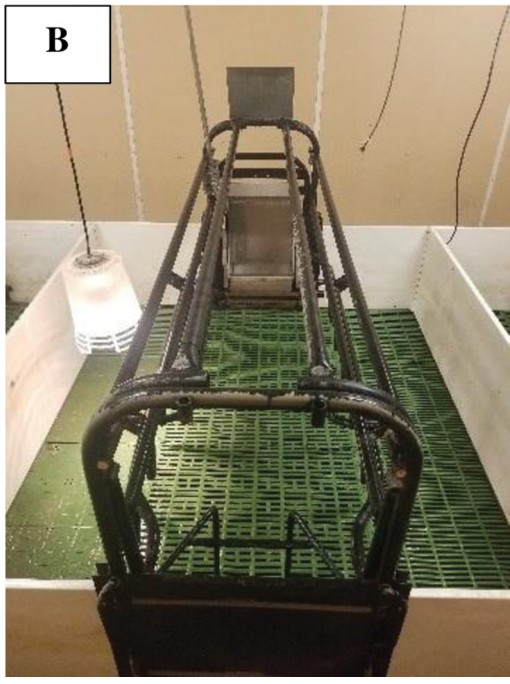

**Fig 1.** Farrowing stall with treatment A) Baby Pig Heat Mat and B) Poly Heat Lamp Fixture used in study comparing the time needed after farrowing ≥ 50% of piglets to locate and lie on- or under the supplemental heat source for ≥ 5 min after the birth of the last-born pig (courtesy of Lane et al., [9, 10]).

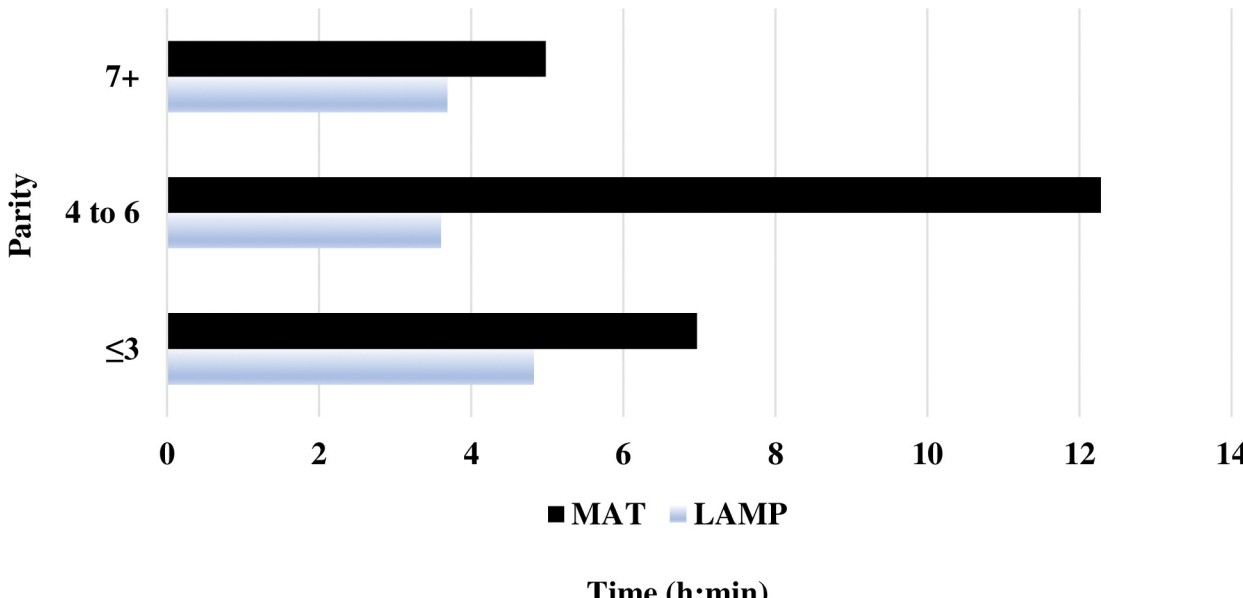

**Fig 2. Descriptive statistics by sow parity on how long it took (h:min) for ≥ 50% of piglets to locate and lie[1] on- or under the supplemental heat[2,3] source for ≥ 5 min after the birth of the last-born pig.** [1]Lying was defined as either sternal or lateral recumbency with ≥ 75% of their body touching the heat mat or inside the lamp heat circle. [2]Baby Pig Heat Mat—Single 48 34 cm W x 122 cm L with a heating area of 0.4 m², polyethylene (Kane Manufacturing, Pleasant Hill, IA). [3]Poly Heat Lamp Fixture 25 cm L x 30 cm W with a heating area of 0.3 m², polypropylene; (HogSlat, Newton Grove, NC) with a 125 W Infrared Heat Bulb (QC Supply, Ames, IA).

parity, number of piglets born alive, and heat source type that related to how long it took for ≥ 50% of piglets to locate and lie on- or under the supplemental heat source for ≥ 5 min after the birth of the last-born pig (Fig 2).

The piglet cohorts laid on- or under the heat source for ~29 mins LAMP piglets and ~42 mins for MAT piglets, respectively (Table 1).

When breaking down the first lying event by sow parity and heat source, piglets assigned the MAT tended to spend more time lying than those piglets assigned to the LAMP (Fig 3).

Average pre-weaning mortality was 11% (LAMP) and 18% (MAT). Piglets assigned the MAT weaned at a heavier average litter weaning weight (4 kg heavier), however, more piglets were weaned from MAT treatment and this could explain the difference (~48 piglets [LAMP] vs. ~52 piglets [MAT]). The MAT heat source used less energy than the LAMP (16 vs. 63 kWh) over the study duration (Table 2).

## Discussion

Piglets are naturally motivated to stay near their dam and littermates several days after farrowing to receive protection, warmth, and nutrition [15–18]. Pigs are attracted to their dam's udder immediately following birth [19]. This attraction to the udder is driven by olfactory cues with piglets having a highly developed sense of smell [20, 21]. Other attractors towards the sow udder include both tactile and thermal properties [16, 19].

Additional heat sources are designed to entice piglets away from the sow, to reduce crushing but provide the necessary warmth. It is important to provide a practical on-farm ABM for caretakers so that they know how quickly the majority of piglets should be moving towards and resting under this heat source after birth. This novel study presents a new practical ABM. It should be noted, that this ABM was not designed to determine the first- and length of time an individual piglet "first" used the heat source. Rather it was designed for a caretaker to be

**Table 1. Descriptive results for first time to locate, lie down, and stay on a heat lamp or heat mat for crossbred piglets.**

| Sow[a] | Parity | Total born | Born alive | Half litter[b] | Total farrowing (h:min)[c] | Find (h:min)[d] | Laid down (min:sec)[e] |
|---|---|---|---|---|---|---|---|
| **LAMP[f]** | | | | | | | |
| 66 | 4 | 7 | 7 | 4 | 2:02 | 0 | 11.45 |
| 567 | 1 | 13 | 13 | 7 | 1:15 | 9.47 | 49.68 |
| 54 | 5 | 13 | 13 | 7 | 1:22 | 9.61 | 37.68 |
| 435 | 7 | 6 | 6 | 3 | 0:55 | 3.68 | 25.70 |
| 568 | 1 | 10 | 10 | 5 | 4:26 | 0.18 | 23.07 |
| 544 | 4 | 6 | 5 | 3 | 4:37 | 1.20 | 28.52 |
| *Avg* | . | **9** | **9** | **5** | **2:26** | **4.83** | **29.35** |
| *Min* | . | **6** | **5** | **3** | **0:55** | **0.00** | **11.45** |
| *Max* | . | **13** | **13** | **7** | **4:37** | **9.61** | **49.68** |
| **MAT[g]** | | | | | | | |
| 62 | 5 | 16 | 13 | 7 | 3:19 | 19.67 | 39.77 |
| 494 | 7 | 6 | 6 | 3 | 1:54 | 4.98 | 42.08 |
| 575 | 1 | 10 | 10 | 5 | 4:14 | 13.25 | 32.18 |
| 508 | 3 | 17 | 14 | 7 | 2:22 | 7.25 | 37.20 |
| 569 | 1 | 10 | 7 | 4 | 8:14 | 0.41 | 81.23 |
| 118 | 5 | 13 | 13 | 7 | 1:30 | 4.89 | 21.65 |
| *Avg* | . | **12** | **11** | **6** | **3:36** | **8.41** | **42.35** |
| *Min* | . | **6** | **6** | **3** | **1:30** | **0.41** | **21.65** |
| *Max* | . | **17** | **14** | **7** | **8:14** | **19.67** | **81.23** |

[a]A total of 12 crossbred sows were enrolled in the study (parity 1 = 4; 3 = 2; 4 = 2; 5 = 2; and 7 = 2).

[b]50% was defined as either an equal number of piglets for those born alive in litters that had an even number or one extra piglet for those piglets born into litters with an odd number of piglets. The difference between Total Born and Born Alive included piglets that were still born and mummies.

[c]Total farrowing was defined as the amount of time from the birth of the first- to the last piglet in the litter.

[d]How long it took for ≥ 50% of piglets to lie on- or under the supplemental heat source for ≥ 5 min after the birth of the last-born pig. Lying was defined as either sternal or lateral recumbency with ≥75% of their body touching the heat mat or inside the lamp heat circle [9, 10]

[e]How long these piglets laid on- or under the heat source for the first time.

[f]Poly Heat Lamp Fixture 25 cm L x 30 cm W with a heating area of 0.3 m², polypropylene; (HogSlat, Newton Grove, NC) with a 125 W Infrared Heat Bulb (QC Supply, Ames, IA).

[g]Baby Pig Heat Mat—Single 48 34 cm W x 122 cm L with a heating area of 0.4 m², polyethylene (Kane Manufacturing, Pleasant Hill, IA).

All averages were rounded to the nearest whole number.

able to determine when ≥ 50% of piglets to lie on- or under the supplemental heat source for ≥ 5 min after the birth of the last-born pig. This 50% was a starting point, as the team considered this a "majority." However, future work could consider other increments and relate this back to survival, such as 25, 75 or 100% respectively.

This ABM fits in to today's protocols and practices for the newborn piglet. Caretakers on commercial farms do not identify and track the piglet at this early stage of life on an individual level. Thus, creating a cheap, effective and practical on-farm tool could aid caretakers in better time management to assist newborns who are struggling to locate the heat source. Knowing when a sow ends farrowing would be the "starting point" for caretakers to note on the sow card, and before they leave for the day they can make sure on their last walk through that at ≥ 50% of piglets to lie on- or under the supplemental heat source for ≥ 5 min after the birth of the last-born pig. If they locate litters that have not met this ABM then they can duly move piglets onto- or under the heat source and make an additional note on the sow card.

The difference in heat source locating time might be explained by the way these heat sources function. Zhu et al., [15] reported that lamps radiate heat in a top-down pattern,

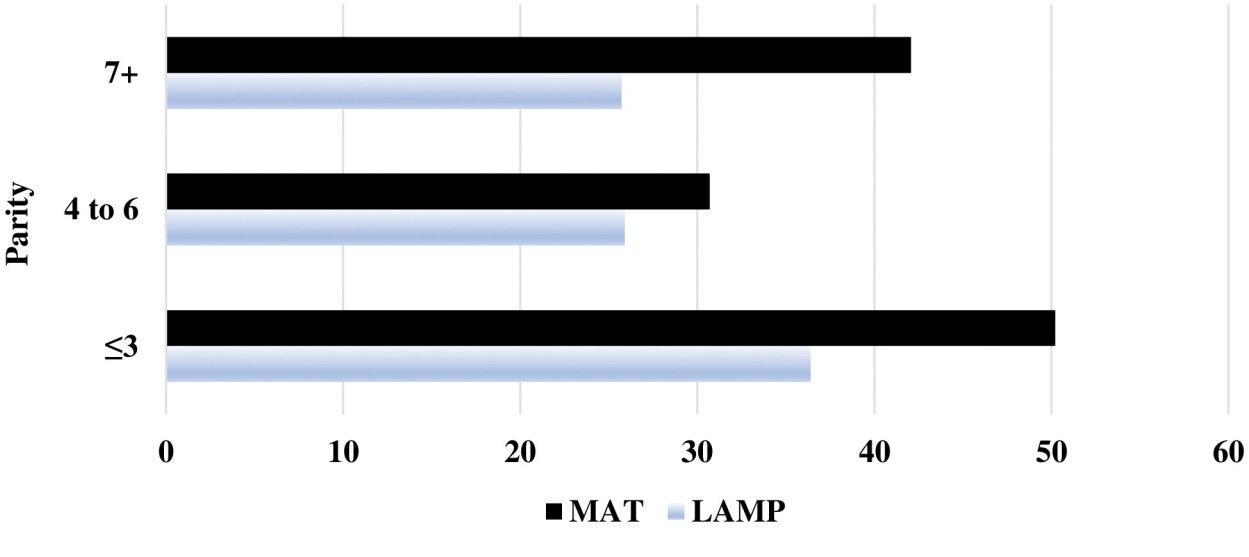

**Fig 3. Descriptive statistics by sow parity on how long (min) piglets stayed lying[1] down on- or under the supplemental heat[2,3] source.** [1]Lying was defined as either sternal or lateral recumbency with $\geq$ 75% of their body touching the heat mat or inside the lamp heat circle. [2]Baby Pig Heat Mat—Single 48 34 cm W x 122 cm L with a heating area of 0.4 m$^2$, polyethylene (Kane Manufacturing, Pleasant Hill, IA). [3]Poly Heat Lamp Fixture 25 cm L x 30 cm W with a heating area of 0.3 m$^2$, polypropylene; (HogSlat, Newton Grove, NC) with a 125 W Infrared Heat Bulb (QC Supply, Ames, IA).

which warms standing piglets better than lying piglets. Davis and others [13] reported that mats provide a more evenly distributed surface temperature and are considered bottom-up heating with their function to heat piglets through their stomachs, which may increase the desire to lie down. At farrowing when piglets are active and standing to perform teat seeking and suckling, they may feel the warming benefits of a lamp quicker over a mat and, in-turn, locate and use it more quickly, which aligns with our study findings. There was not a clear pattern related to born alive influencing the time for $\geq$ 50% of piglets to locate and lie on- or under the supplemental heat source for $\geq$ 5 min after the birth of the last-born pig. Furthermore, anecdotal evidence noted that for both treatment groups, piglets moved from the heat source to nursing, and this usage time corresponded to previous work on nursing intervals which ranged between 35 and 40 min [22].

Heat from a lamp is concentrated right under the bulb and heating intensity decreases as you move out towards the periphery. Heat from lamps could contribute to sow heat stress and discomfort, which might increase piglet crushing deaths. National U.S. pre-weaning mortality is 14.4% [23], and in this study, pre-weaning for LAMP piglets was lower at 11% compared to MAT piglets at 18%. An interesting study by Christison et al., [24] reported that when caretakers moved piglets to the supplemental heat, liveability improved from 79 to 94%. Therefore, pre-weaning mortality may have improved if we had implemented this best practice policy, but it would have interfered with the behavioral measures we were striving to collect.

In conclusion, to the authors' knowledge, this is the first published study using a continuous sampling method to precisely examine a new ABM measure and to determine how long this cohort stayed. Our findings show an immense time range in locating and lying under- or on the heat source. Therefore, we suggest that caretakers should be trained to use this ABM and assist all piglets to locate the heat source after farrowing is complete to improve piglet liveability.

**Table 2. Descriptive results for litter performance and energy usage when given a heat lamp or heat mat.**

| Sow[a] | Parity | Pre-wean mortality (%) | Litter birth weight (kg) | Litter wean weight (kg) | Energy kWh |
|---|---|---|---|---|---|
| **LAMP[b]** | | | | | |
| 66 | 4 | 00.00 | 13.53 | 45.88 | 54.93 |
| 567 | 1 | 7.69 | 15.90 | 56.12 | 55.01 |
| 54 | 5 | 00.00 | 17.59 | 76.28 | 69.42 |
| 435 | 7 | 16.16 | 8.91 | 33.27 | 70.76 |
| 568 | 1 | 20.00 | 11.14 | 33.80 | 64.49 |
| 544 | 4 | 20.00 | 7.25 | 29.25 | 64.35 |
| *Avg* | . | **11** | **12** | **46** | **63** |
| *Min* | . | **00.00** | **7.25** | **29.25** | **54.93** |
| *Max* | . | **20.00** | **17.59** | **76.28** | **70.76** |
| **MAT[c]** | | | | | |
| 62 | 5 | 38.46 | 21.43 | 71.53 | 21.19 |
| 494 | 7 | 00.00 | 10.00 | 42.62 | 19.19 |
| 575 | 1 | 00.00 | 12.21 | 45.44 | 19.74 |
| 508 | 3 | 7.14 | 16.01 | 58.27 | 5.86 |
| 569 | 1 | 57.14 | 8.21 | 10.83 | 12.43 |
| 118 | 5 | 7.69 | 18.74 | 72.86 | 17.07 |
| *Avg* | | **18** | **14** | **50** | **16** |
| *Min* | | **00.00** | **8.21** | **42.62** | **5.86** |
| *Max* | | **57.14** | **21.43** | **72.86** | **21.19** |

[a]A total of 12 crossbred sows were enrolled in the study.

[b]Poly Heat Lamp Fixture 25 cm L x 30 cm W with a heating area of 0.3 m$^2$, polypropylene; (HogSlat, Newton Grove, NC) with a 125 W Infrared Heat Bulb (QC Supply, Ames, IA).

[c]Baby Pig Heat Mat—Single 48 34 cm W x 122 cm L with a heating area of 0.4 m$^2$, polyethylene (Kane Manufacturing, Pleasant Hill, IA).

All averages were rounded to the nearest whole number.

## Acknowledgments

A special thanks to the farm staff at the Allen E. Christian Iowa State University Swine Teaching farm for their assistance in completing this project.

## Author Contributions

**Conceptualization:** Anna K. Johnson, Kenneth J. Stalder.

**Data curation:** Cassandra R. Stambuk.

**Formal analysis:** Kenneth J. Stalder.

**Funding acquisition:** Anna K. Johnson.

**Investigation:** Anna K. Johnson, Kenneth J. Stalder.

**Methodology:** Cassandra R. Stambuk, Anna K. Johnson, Kenneth J. Stalder.

**Project administration:** Anna K. Johnson, Kenneth J. Stalder.

**Resources:** Anna K. Johnson, Kenneth J. Stalder.

**Supervision:** Anna K. Johnson, Kenneth J. Stalder.

**Writing – original draft:** Cassandra R. Stambuk, Anna K. Johnson, Karli J. Lane, Kenneth J. Stalder.

**Writing – review & editing:** Cassandra R. Stambuk, Anna K. Johnson, Karli J. Lane, Kenneth J. Stalder.

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
