## [Decision Letter · Decision Letter 0]

23 Aug 2023

PONE-D-23-21093From Birth to Lying on- or Under a Supplemental Heat Source: How Long does it take for Half the Litter to Lie Down and How Long Do They Stay?PLOS ONE

Dear Dr. Johnson,

Thank you for submitting your manuscript to PLOS ONE. After careful consideration, we feel that it has merit but does not fully meet PLOS ONE’s publication criteria as it currently stands. Therefore, we invite you to submit a revised version of the manuscript that addresses the points raised during the review process.

We look forward to receiving your revised manuscript.

Kind regards,

Ewa Tomaszewska, DVM Ph.D

Academic Editor

PLOS ONE

“This project was funded in part by Iowa Farm Bureau Federation (AJ) and Kane Manufacturing (KS). Partial funding of Dr. Johnson’s and Dr. Stalder’s salary is supported by the Department of Animal Science, College of Agriculture and Life Sciences at Iowa State University, and the US Department of Agriculture.”

“This project was funded in part by Iowa Farm Bureau Federation and Kane Manufacturing. Partial funding of Dr. Johnson’s and Dr. Stalder’s salary is supported by the Department of Animal Science, College of Agriculture and Life Sciences at Iowa State University, and the US Department of Agriculture.”

“This project was funded in part by Iowa Farm Bureau Federation (AJ) and Kane Manufacturing (KS). Partial funding of Dr. Johnson’s and Dr. Stalder’s salary is supported by the Department of Animal Science, College of Agriculture and Life Sciences at Iowa State University, and the US Department of Agriculture.”

6. Please include a separate caption for each figure in your manuscript.

Reviewers' comments:

Reviewer's Responses to Questions

**Comments to the Author**

1. Is the manuscript technically sound, and do the data support the conclusions?

Reviewer #1: Yes

2. Has the statistical analysis been performed appropriately and rigorously? 

Reviewer #1: N/A

3. Have the authors made all data underlying the findings in their manuscript fully available?

Reviewer #1: Yes

4. Is the manuscript presented in an intelligible fashion and written in standard English?

Reviewer #1: Yes

5. Review Comments to the Author

Reviewer #1: This is a very nice study looking at a very simple practice that can lead to improved caretaker tools and observation practices. Some comments are provided below to clarify information about the purpose and background of the goals of this manuscript.

Line 54: Remove the dash after 250. In the second sentence of line 54, please remove “, respectively”.

Lines 59-61: Curious as to why the authors were only interested in looking at half the litter as the minimum point. Can the rationale be explained why at least half the litter was the target for this research? Why not the full litter? Or 75% or some other proportion?

Line 70: Remove the comma after “Location”

Line 76: Please rephrase: "...creep area, where the heat source provided measured 1.2 m..."

Line 78: Remove the parenthesis at end of reference.

Line 116, last sentence: Why were no statistical analyses completed or presented in this manuscript?

Line 119-120: Was parity evenly distributed between treatments?

Line 127: Please rephrase as: "When breaking down the first lying event by sow parity and heat source,..."

Line 128: where it is stated “MAT piglets”, please remove the word piglets here...it is already stated earlier in the sentence.

Lines 137-140: Please provide more discussion on how newborn piglets perceive their environment, particularly to seeking warmth and needs for thermoregulation.

Lines 140-142: Please provide more context on the lack of ABM for caretakers or currently used ABM that are practical and used in commercial swine settings. Please emphasize why its important for ABMs to be practical and what is needed to provide more tools like these to caretakers.

Lines 142-143: This is the first mention of this study focusing on its goals relative to "a new practical ABM". This should be introduced in the intro section and more detail should be provided here to emphasize how this study's methods yield such an ABM that would be practical at the farm/caretaker-level.

Line 143: Please begin sentence with "On the farm".

Lines 144-147: Please rephrase this last section of the sentence. It is difficult to follow.

Lines 150-153: Do the authors believe that although it took piglets longer to find the mats, this may be why they chose to stay on the mat longer?

Line 154: Remove comma after "seeking"

Line 165: Please move the comma to after the reference.

Lines 166-168: Are there any studies where preference testing of heat sources by piglets was conducted?

Line 166: Check formatting for references, see a bracket and parenthesis here.

Line 175: Please rephrase as: "should be trained to use this ABM..."

Table 1, line 252 (Footnote): "...born alive IN litters that..." Missing word "in"

Table 2: Please include a header/title for the last column of the table.

Figure 1, line 2 (Figure description): Add the word "for" after farrowing. "...after farrowing FOR ..."

Figure 1, line 4 (Figure description): Add parenthesis at end of reference.

Figure 2 graph: Please relabel your x-axis as "Time (h:min)"

Figure 3 graph: Please center your axis label below the treatments and relabel as "Time (min)"

6. PLOS authors have the option to publish the peer review history of their article (what does this mean?). If published, this will include your full peer review and any attached files.

Reviewer #1: **Yes: **Michelle Calvo-Lorenzo

---

## [Author Response · Author response to Decision Letter 0]

29 Sep 2023

Good afternoon,

per Sloan Santos direction I have updated the data availability statement in the author comments section of the online submission. Sloan noted "I will update tour data availability statement on your behalf." Thank you A.

---

## [Decision Letter · Decision Letter 1]

11 Oct 2023

PONE-D-23-21093R1From Birth to Lying on- or Under a Supplemental Heat Source: How Long does it take for Half the Litter to Lie Down and How Long Do They Stay?PLOS ONE

Dear Dr. Johnson,

Thank you for submitting your manuscript to PLOS ONE. After careful consideration, we feel that it has merit but does not fully meet PLOS ONE’s publication criteria as it currently stands. Therefore, we invite you to submit a revised version of the manuscript that addresses the points raised during the review process.

We look forward to receiving your revised manuscript.

Kind regards,

Ewa Tomaszewska, DVM Ph.D

Academic Editor

PLOS ONE

Journal Requirements:

Reviewers' comments:

Reviewer's Responses to Questions

**Comments to the Author**

1. If the authors have adequately addressed your comments raised in a previous round of review and you feel that this manuscript is now acceptable for publication, you may indicate that here to bypass the “Comments to the Author” section, enter your conflict of interest statement in the “Confidential to Editor” section, and submit your "Accept" recommendation.

Reviewer #1: All comments have been addressed

2. Is the manuscript technically sound, and do the data support the conclusions?

Reviewer #1: Yes

3. Has the statistical analysis been performed appropriately and rigorously? 

Reviewer #1: N/A

4. Have the authors made all data underlying the findings in their manuscript fully available?

Reviewer #1: Yes

5. Is the manuscript presented in an intelligible fashion and written in standard English?

Reviewer #1: Yes

6. Review Comments to the Author

Reviewer #1: The authors addressed all my questions and comments, and the paper contains more clarity in its methods, purpose, and overall conclusions. I only have one more comment for the authors: Given that this is being submitted as a research paper and data is only presented descriptively, the authors should explain why the data set and analysis are presented descriptively (i.e., that no stats analysis was conducted and why).

7. PLOS authors have the option to publish the peer review history of their article (what does this mean?). If published, this will include your full peer review and any attached files.

Reviewer #1: No

---

## [Author Response · Author response to Decision Letter 1]

19 Oct 2023

Citations have been checked - done

I have attended to the reviewers comment (stats vs. descriptive) - done

Details can be located in the newly uploaded cover letter

thank you A.

---

## [Decision Letter · Decision Letter 2]

31 Oct 2023

From Birth to Lying on- or Under a Supplemental Heat Source: How Long does it take for Half the Litter to Lie Down and How Long Do They Stay?

PONE-D-23-21093R2

Dear Dr. Anna Kerr Johnson,

We’re pleased to inform you that your manuscript has been judged scientifically suitable for publication and will be formally accepted for publication once it meets all outstanding technical requirements.

Kind regards,

Ewa Tomaszewska, DVM Ph.D

Academic Editor

PLOS ONE

Additional Editor Comments (optional):

Reviewers' comments:

Reviewer's Responses to Questions

**Comments to the Author**

1. If the authors have adequately addressed your comments raised in a previous round of review and you feel that this manuscript is now acceptable for publication, you may indicate that here to bypass the “Comments to the Author” section, enter your conflict of interest statement in the “Confidential to Editor” section, and submit your "Accept" recommendation.

Reviewer #1: All comments have been addressed

2. Is the manuscript technically sound, and do the data support the conclusions?

Reviewer #1: Yes

3. Has the statistical analysis been performed appropriately and rigorously? 

Reviewer #1: Yes

4. Have the authors made all data underlying the findings in their manuscript fully available?

Reviewer #1: Yes

5. Is the manuscript presented in an intelligible fashion and written in standard English?

Reviewer #1: Yes

6. Review Comments to the Author

Reviewer #1: The authors have addressed all my questions and comments, and the paper contains more clarity in its methods, purpose, and overall conclusions. I appreciate the authors' response on presenting the data in a descriptive form (given that the variation in sows were so high) as the correct tool for biological interpretation of each individual sow/litter unit. This paper has a lot of practical and implementable information that will be a great tool for improving swine welfare in commercial operations.

7. PLOS authors have the option to publish the peer review history of their article (what does this mean?). If published, this will include your full peer review and any attached files.

Reviewer #1: No

---

## [Editor Report · Acceptance letter]

11 Dec 2023

PONE-D-23-21093R2 

From Birth to Lying on- or Under a Supplemental Heat Source: How Long does it take for Half the Litter to Lie Down and How Long Do They Stay? 

Dear Dr. Johnson:

I'm pleased to inform you that your manuscript has been deemed suitable for publication in PLOS ONE. Congratulations! Your manuscript is now with our production department. 

Kind regards, 

on behalf of

Professor Ewa Tomaszewska 

Academic Editor

PLOS ONE